# Effects of Restorative Environment and Presence on Anxiety and Depression Based on Interactive Virtual Reality Scenarios [note 1]

**DOI:** 10.3390/ijerph19137878

**Published:** 2022-06-27

**Authors:** Zhimeng Wang, Yue Li, Jingchen An, Wenyi Dong, Hongqidi Li, Huirui Ma, Junhui Wang, Jianping Wu, Ting Jiang, Guangxin Wang

**Affiliations:** 1Department of Psychology, School of Humanities and Social Sciences, Beijing Forestry University, Beijing 100083, China; 202028061042@mail.bnu.edu.cn (Z.W.); yu4712li-s@student.lu.se (Y.L.); dwypsyche@outlook.com (W.D.); lihongqidi@bjfu.edu.cn (H.L.); mahuirui@bjfu.edu.cn (H.M.); wjhdexinyang@163.com (J.W.); wgx8868@163.com (G.W.); 2Faculty of Psychology, Beijing Normal University, Beijing 100875, China; 202028061001@mail.bnu.edu.cn (J.A.); psytingjiang@bnu.edu.cn (T.J.); 3Department of Psychology, Lund University, 22100 Lund, Sweden

**Keywords:** virtual reality, restorative environment, presence, anxiety, depression, interaction

## Abstract

Anxiety and depression have been growing global mental health problems. The following studies explored the effect of interactive VR scenarios to find a low-cost and high-efficiency solution. Study 1 designed a 2 (anxiety and depression state) × 4 (interactive VR scenarios) experiment, the results of 20 participants showed that the designed scenarios had good restoration and presence, assisting to improve depression mood for people with mild to moderate anxiety and depression. Study 2 further investigated the intervention effects of two environment types (urban and park) and four interactive activities (automatic viewing, free-roaming, fishing, and watering plants in the park environment), based on data from a 10-minute experiment conducted by 195 participants with mild to moderate anxiety and depression. The subjective scales, EEG and EMG, and scenario experience were analyzed and the results showed that: (1) the restorative and present VR scenarios were beneficial in alleviating state anxiety and depression; (2) the restorative environment and presence were significantly and positively related to the reduction of anxiety and depression respectively, moreover, presence mediated the restorative environment on the recovery from anxiety and depression; (3) the environmental settings, the complexity of interaction, human factors, and maturity of VR devices and technology were also key factors that influenced the effects of interactive VR scenario experience and intervention. These studies revealed VR psychological intervention scenarios could be designed with comprehensive factors. Moreover, they might help pave the way for future study in exploring the physiology and psychology mode in virtual and real spaces, enhancing intervention effectiveness.

## 1. Introduction

The negative effects of anxiety and depression on people’s physical and mental health have created a significant medical and financial burden on society in recent years [1,2,3]. In particular, anxiety and depression were generally elevated among the public during the COVID-19 pandemic, with the prevalence of anxiety disorders and major depression increasing by 25.6% and 27.6% respectively [4]. Meanwhile, suicidal ideation and suicidal behavior occurred in vulnerable groups, highlighting the seriousness of public health problems [5,6,7]. Currently, the main methods for relieving anxiety and depression are psychological interventions, medicinal treatments, or a combination of both [8]. Among them, psychological interventions have advantages in enhancing positive emotions and correcting cognitive biases [9,10]. However, traditional psychological interventions are challenged by poorly established psychological service systems, unaffordable service costs, patient stigma, and mistrust [11,12,13]. Finding low-cost, efficient, and flexible solutions to psychological problems for the public has become one of the core tasks of mental health and its interdisciplinary field [14,15,16].

New technologies represented by virtual reality, artificial intelligence, and biofeedback have brought positive energy to the mental health field with the advent of the intelligence era [17]. In particular, virtual reality has great potential for psychological interventions because of its immersion, interactivity, imagination, and safety. Immersive virtual reality systems, naturally stimulating human perceptual experiences through interactive tools (e.g., helmet displays, bodysuits, and data gloves) [18,19], are widely used in the assessment and intervention of mental health problems [20,21,22,23,24]. The effective VR interventions for anxiety and depression are VR exposure-based cognitive behavioral therapy [25], VR biofeedback [26], and VR exercise [27]. However, current research focuses on mental disorders, ignoring the growing public anxiety and depression emotion. VR might be a very effective tool for early intervention [25], and the feasibility, as well as the effectiveness of VR-based interventions for mild to moderate emotional problems, need to be revealed. In addition, the factors and mechanisms influencing VR-based interventions have not been clearly identified.

The symbiotic relationship between humans and the environment plays an important role in the development of public mental health. Research that focuses on the ecological models in mental health (from the micro-individual to the macro-environment), exploring the positive energy from the environment, has generated extensive discussion and continues to grow [28,29,30]. Among them, restorative environment studies point to the natural advantages of ecological environments in promoting mental health. A restorative environment is one that facilitates the recovery of physical and mental resources depleted in the development process, not only for general energy but also for abnormal states and long-term well-being [31,32,33,34]. Stress Recovery Theory (SRT) and Attention Recovery Theory (ART) reveal positive physiological and psychological connections between humans and nature from emotional and cognitive perspectives respectively. SRT suggests the human aesthetic need for a natural environment activates the parasympathetic nervous system, evoking positive emotions and releasing stress [35,36]. ART argues that directed attention needs to be used to capture specific information which is the cornerstone of human cognitive function, and restorative environments can help alleviate discomforts caused by excessive directed attention and even invoke reflection and insight into life. Moreover, restorative environments feature being away (different from the everyday environments which could rest directed attention), extent (rich, coherent, and sufficient settings), fascination (attractive and no will effort required), and compatibility (matched with individual inclinations) characteristics [32,33,37,38,39,40]. The natural environments are representative of the restorative environments. Researchers have found that exposure to natural environments could reduce people’s blood pressure, heart rate variability, skin electricity, and cortisol levels and increase positive emotions and well-being, enhance executive function and self-regulation, and improve anxiety and depression [41,42,43,44,45,46].

The combination of restorative environments and VR creates new opportunities for recovery from anxiety and depression. VR-based restorative environments provide natural environmental settings (e.g., forests, vegetation, the sound of birds and running water) [47,48] and interactive spaces (e.g., aerobic exercise and rehabilitation training) [49,50], which could promote parasympathetic activity and make people feel safe and relaxed, effectively relieving physical and mental stress and improving mental health [51,52,53]. However, fewer studies report on environmental settings outside of forests and interactive VR restorative scenarios, how to design a VR restorative environment with good experience, and the mechanisms of VR restorative environments on anxiety and depression are unclear.

Physiological indicators play an important role in reflecting the feasibility as well as the mechanisms of VR interventions for anxiety and depression. It has been shown that tracking electroencephalography (EEG) in VR scenarios could capture the levels of alertness, calmness, and engagement, helping understand the physical and mental state, and offers the potential for individualized treatment [54,55,56]. EEG is a regular electrical signal detected during neuronal activation in the brain and is classified as δ (0.5–4 Hz), θ (4–8 Hz), α (8–13 Hz), β (13–30 Hz), and other major waveforms according to different frequencies. Generally, α waves are mainly associated with wakefulness, eye closure, and relaxation; β waves reflect emotional and cognitive processes and are related to sensorimotor behavior [57,58]. The ratio of relative power spectral densities between α and β was a sensitive indicator reflecting brain activity. For example, α/β, which shows the level of calmness and alertness, is used in the field of human–computer interaction [59] and cognitive psychology [60]. Furthermore, the ratio of α/β has been proved to detect the fatigue level of human beings [61]. In addition, surface electromyography (EMG) is calculated to capture muscle loading in interactive activities. Moreover, the median frequency (MF) is frequently used to evaluate muscle fatigue in both healthy groups and in patient groups [62]. It has been shown that MF is a non-invasive and sensitive physiological measure, reflecting muscle activation and movement during cognitive and behavioral activities [63,64,65]. Therefore, the α/β and MF might be available to reflect people’s emotional and cognitive states in interactive VR scenarios, facilitating the optimization of the human–computer interaction experience.

Interactive VR scenarios can be developed with 360${}^{\circ }$ panoramic cameras or based on 3D modeling software. The modeling-based VR offers closer scenarios that can be not only viewed but also interacted with in a natural and controlled state, just like real-world activities, or what could not happen in real life becomes a reality [66,67]. People could interact with the virtual space through gaze interaction, gesture interaction, interactive devices, etc. [68]. At the same time, people’s movement and proprioception are closely linked, and the perception of self and environment is enhanced, arising from the presence and positive emotions [69]. Presence is a psychological experience of feeling the existence of the self in a virtual space, although the body is in the real world, and is divided into subjective presence (self-perception), social presence (interaction with other people), and environmental presence (interaction with the environment) depending on the object of interaction [70,71,72], containing the place illusion (being there) and the plausibility (the events seem really happening) [19,73,74]. Presence differs from immersion in that immersion focuses on the realism and naturalness of the simulated senses (vision, auditory, olfaction, taste, tactus, and other sensory channels) from a technical point of view, requiring a high degree of simulation and resolution, which are limited by technical and economic conditions, but presence describes a perception that is separate from the physical world and the participatory states of everyday life, creating or even altering conscious experience [75,76,77,78]. The Immersion, Presence, Performance (IPP) model states that immersion is a prerequisite for presence [79]. However, it is not only realistic and immersive environments that can inspire presence, but scenarios that are appropriate to individual performance, with rich and vivid extent (settings with breadth and depth) and good interactivity (naturalness and ease of engaging in interactive activities) [75,80,81]. Presence might play an important role in VR interventions; viewing VR natural scenarios creates a sense of presence, which can be beneficial in reducing state anxiety and enhancing positive emotions [53,82]. However, there are fewer studies on the effects of VR presence and interventions, and whether enhancing VR extent and interactivity is beneficial in promoting presence to improve anxiety and depression still needs to be further explored [83].

In summary, to explore the effects and mechanisms of a restorative environment and presence on anxiety and depression based on interactive VR scenarios and improve the experience and the effectiveness of the VR-based intervention, these studies were divided into two parts: Study 1 designed and developed a virtual park with interactive scenarios based on SRT and ART, and initially validated its restorative effects of the people with mild to moderate anxiety and depression mood states through experimental methods; Study 2 compared the effects of various environment types and interactive activities on VR experience and intervention effects at the physiological and psychological levels through quantitative and qualitative research methods to reveal the interrelationships between restorative environments, presence, changes in anxiety, and changes in depression.

## 2. Study 1: Design and Initial Testing of Restorative and Present VR Scenarios

### 2.1. Method

#### 2.1.1. Participants

Participants were recruited through posters and informed of possible risks and privacy protection regulations. A total of 26 undergraduate students (N ${}_{male=5}$) from Beijing Forestry University participated in the study, with an average age of 19.58, normal or corrected vision, and no major physical or mental illness. After excluding missing information, complete data from 20 individuals were included in the analysis. They were divided into the mild to moderate anxiety and depression group (Group 1, whose scores on the State Anxiety, Trait Anxiety and Depression scales were all greater than or equal to 40 and less than or equal to 60) and the other group those who were in mild anxiety or mild depression (Group 2). There were 12 participants (N ${}_{male=3}$) in the first group with a mean age of 19.75 and 8 participants (N ${}_{male=1}$) in the second group with a mean age of 18.63. Participants were paid CNY 20 for completing the experiment. The experiment was approved by the Ethics Committee of the Department of Psychology, Beijing Forestry University.

#### 2.1.2. Experimental Design

A 2 (anxiety and depression state) × 4 (interactive VR scenarios) experimental design was used. The anxiety and depression state was the between-subjects variable, divided into Group 1 (the mild to moderate anxiety and depression group) and Group 2 (the other). The interactive VR scenarios were the within-subjects variable, including flying a kite in the lawn area (Act 1), watering vegetables in the gardening area (Act 2), fishing in the water area (Act 3), and feeding birds in the forest area (Act 4). A completely randomized design was used to control the effects of order effects, experimental environment, and individual differences. Specifically, to ensure that the VR scenarios for the current experiment were presented in a completely random order, scenario selection was based on the built-in python command for random integer generation.

#### 2.1.3. Materials and Instruments

##### Interactive VR Scenarios

Concerning the design of interactive VR scenarios, based on the previous theories and research on restorative environments and virtual reality, and with reference to environmental psychology, landscape design, rehabilitation medicine, and other related materials, this study created a virtual park by selecting suitable environmental elements, mainly containing four environmental areas, such as lawn area, horticultural planting area, water area, and forest area. Each area was independent and interconnected, with a preference for achieving various functions: finding an open space to relax and flying a kite in the lawn area; enjoying the idyllic scenery and watering vegetables in the horticultural planting area; feeling the rhythm of the water body and fishing in the water area; viewing green plants and feeding birds in the forest area, as shown in Figure 1.

Concerning the technical implementation of interactive VR scenarios, software such as PS CC2019 (Adobe Systems Inc., SanJose, CA, USA), AI CC2019 (Adobe Systems Inc., San Jose, CA, USA), CAD 2019 (Autodesk, Inc., San Rafael, CA, USA), and SketchUp 2019 (Trimble Inc., Sunnyvale, CA, USA) were used to design and test the feasibility of technical implementation in the pre-construction phase. The environment elements, paths, and colors were evaluated and tested by a professor of environmental psychology, an associate professor of art and design, and a master of landscape architecture after the preliminary scenarios were presented using Lumion 8.0 software (Act-3D B. V., Sassenheim, The Netherlands). The immersive VR scenarios were developed through Unity 5.6.4 (Unity Technologies Inc., San Francisco, CA, USA) in the later stages. Users could freely move and complete activities through the handles. However, the space available for the move was limited in order to avoid collisions or other discomforts.

##### Subjective Scales

The Restoration Environment Scale (RES) was used to assess the restoration of the design scenarios. The RES is the first Chinese version of the Restoration Environment Scale developed by Liuhong Ye, Fan Zhang, and Jianping Wu based on ART theory in 2010. It is a 22-item questionnaire, scored on a 7-point scale, which includes three factors: being away, fascination and compatibility, and extent. The Cronbach’s alpha coefficients for the total scale and the three subscales ranged from 0.769–0.936, and the split-half reliability distribution was 0.695–0.903, which shows good reliability and validity.

The Presence Questionnaire (PQ) was used to assess the sense of presence in virtual scenarios, which was revised in 2005 by Bob G. Witmer, Christian J. Jerome, and Michael J. Singer. This 29-item questionnaire is scored on a 7-point scale that includes four factors: involvement, sensory fidelity, adaptation/immersion, and interface quality. It has good reliability and validity.

The State-Trait Anxiety Inventory (STAI) was used to assess participants’ anxiety mood. It was developed and revised by Charles Spielberger et al. The 40-item scale is rated on a scale of 4-point. Items 1–20 are the State Anxiety Inventory (SAI), and items 21–40 are the Trait Anxiety Inventory (TAI). The former describes an unpleasant short-term emotional experience such as tension, fear, apprehension, etc., often accompanied by hyperfunction of the vegetative nervous system. The latter is used to describe a relatively stable tendency to anxiety as a personality trait with individual differences.

The Self-rating Depression Scale (SDS) was used to assess participants’ depression moods. It was developed by Willian W.K. Zung in 1965, containing 20 items on a 4-point scale, which provides a fairly visual representation of the symptoms associated with depressive states and their severity and variability.

##### Instruments

The VR equipment was a set of HTC Vive pro eye, including 1 headset, 2 handles, 2 version 2.0 positioners, and their data connection cables.

The computer device was a set of Alien assembly machines (CPU: Intel Core i7 5820K; GPU graphics card: NVIDIA${}^{®}$ GeForce${}^{®}$ RTX 2080 Ti), which was used to provide optimal environmental definition and smoothness.

#### 2.1.4. Experimental Procedures

Participants experienced the VR intervention once a week and took up to a month to complete, as shown in Figure 2. They signed an informed consent form and completed a pre-test for anxiety and depression, as well as VR adaptive training in the first week. The STAI and SDS were filled out in the pre-test. The operation of the headset and handles was learned in the VR adaptive training. They adjusted the tightness, focal length, and pupillary distance of the headset, to ensure the vision field was clear and blur-free. Moreover, participants would be able to use the handles smoothly, defined as entering the VR scenario and completing the commands “forward, backward, momentary movement and pulling the trigger” with a score of 3 or more each (5-point, assessed by people with sufficient operational experience). In addition, participants would be able to smoothly use handles, which is defined as entering a VR scenario and completing the commands “forward, backward, move left, move right, teleportation and press the trigger”, scoring 3 or more on each item (a 5-point Likert scale, assessed by someone with sufficient operational experience). The participants then experience the VR scenario for 8–10 min once a week for 4 sessions. When the experience was complete, they filled in the RES and PQ if they were in a good state, or after a break if they felt motion sickness. At the fourth week, participants completed the post-test for anxiety and depression and were invited to participate in a VR scenario experience interview, where they were asked “How did you feel in VR and what impression did each scenario leave on you?”.

### 2.2. Results

#### 2.2.1. Restorative Environment

SPSS 25 was used for statistical analysis, and the Kolmogorov–Smirnov test showed that scores were normally distributed. The scores of the restorative environment in group1 and group2 are shown in Figure 3. Repeated-measures ANOVA was used to compare restoration in the four experiments. After chi-squared and sphericity tests, there was no significant difference in the restorative environment scores between the two groups, *F(1,16) = 0.989, p = 0.335*. There was no significant difference in the restorative environment scores within groups, *F(3,48) = 0.383, p = 0.766*.

#### 2.2.2. Presence

The presence scores of Group 1 and Group 2 are shown in Figure 4. Repeated measures ANOVA was used to compare the presence in the four experiments. After chi-squared and sphericity tests, there was no significant difference in presence scores between the two groups, *F(1,16) = 0.546, p = 0.471*. There was no significant difference in the sense of presence scores within groups, *F(3,48) = 0.389, p = 0.761*.

#### 2.2.3. Anxiety and Depression

Descriptive statistics of the pre-test ($SAI_{0}$, $SDS_{0}$, and $TAI_{0}$ in the first experiment) and post-test ($SAI_{1}$, $SDS_{1}$, and $TAI_{1}$ in the fourth experiment) on the anxiety and depression scores for group1 and group2 are shown in Figure 5. Paired *t*-test was used to compare the difference between the pre-test and the post-test. As for Group 1, there was no significant difference between $SAI_{0}$ and $SAI_{1}$, *t(11) = 0.573, p = 0.578*; there was no significant difference between $TAI_{0}$ and $TAI_{1}$, *t(11) = −0.062, p = 0.952*; there was significant difference between $SDS_{0}$ and $SDS_{1}$, *t(11) = 2.799, p = 0.017*. As for Group 2, there was significant difference between $SAI_{0}$ and $SAI_{1}$, *t(7) = −2.638, p = 0.033*; there was no significant difference between $TAI_{0}$ and $TAI_{1}$, *t(7) = −1.986, p = 0.087*; there was no significant difference between $SDS_{0}$ and $SDS_{1}$, *t(7) = −2.308, p = 0.054*.

#### 2.2.4. Scenario Experience

Interviews revealed the overall experience in the virtual park and the experience in each interactive scenario. The simple coding of interview data is shown in Appendix A, Table A1. Naturalness, immersion, relaxation, and enjoyment were the experience of the VR scenarios, which linked to the scenario elements, size, spatial position, colors, the quality of the interface, the interaction, guide, and other factors.

### 2.3. Discussion

The interactive VR scenarios designed in this study had high scores for the restorative environment and presence, allowing participants to forget reality and experience the pleasure of virtual scenarios. However, the mild to moderate anxiety and depression group rated the restorative environment higher than the other group in all scenarios, but had a lower sense of presence scores, suggesting that participants in the two groups might have different preferences for the environment and interaction. Interviews revealed that participants had a better experience with watering vegetables in the horticultural planting area and fishing in the water area than flying kites in the lawn area and feeding birds in the forest, possibly because they had good usability and provided timely feedback. However, how interactive VR scenarios affect anxiety and depression still remained unclear, and the type of environments and the complexity of interaction needed to be studied in depth.

Various trends of intervention were revealed. For the mild to moderate anxiety and depression group, there was a significant reduction in depression but not in the state of anxiety. While the other group had no significant post-intervention effect, and even had an increase in anxiety and depression. In addition, there was no significant change in trait anxiety in either group. Studies suggested that anxious and depressive individuals were more sensitive to negative information, showing characteristics such as attentional alerting and difficulty in attention disengaging (getting rid of a threat after attention), which may be related to their experiences of more negative events or the attention preference for negative stimulus [84,85,86,87]. The fresh stimulus in the virtual space, as well as negative stimulus (unsuccessful manipulation or lack of timely feedback), might stimulate innate or acquired fears and enhance specific attentional preferences, leading to very different intervention effects on anxious and depressed moods.

In summary, VR scenarios based on the restorative environment and presence showed significant intervention effects on depression mood for people with mild to moderate anxiety and depression. Watering vegetables in the horticultural planting area and fishing in the water area gained applause. However, the factors affecting the psychological VR intervention were not sufficiently captured. The impact of attention preferences on the interactive scenarios experiences should be noted, including the environment and interaction. On the other hand, scenario experience might expose the secrets of VR intervention which could be deeply analyzed by the qualitative method. Therefore, Study 2 updated the VR scenarios to make clear the effectiveness and mechanisms of the physical and mental effects on anxiety and depression.

## 3. Study 2: The Effect of Restorative and Present VR Scenarios on Anxiety and Depression

### 3.1. Methodology

#### 3.1.1. Participants

A total of 369 students from Beijing Normal University and Beijing Forestry University with normal or corrected vision, good attention, executive ability, and no major physical or mental illness were recruited through posters. Scores of greater than or equal to 40 and less than or equal to 60 on both the SAI and SDS were used as the basis for screening, and 195 participants were randomly assigned to one of the five experimental groups and each was paid CNY 20 at the end of the experiment. After excluding outliers and missing values, data from 186 participants were included in the analysis, and the basic information of participants is shown in Table 1. The experiment had been approved by the ethics committee of the Department of Psychology at Beijing Forestry University.

#### 3.1.2. Experimental Design

A one-way randomized experimental design was used to divide participants into the VR urban environment group (Group 1), VR park environment group (Group 2), VR free-roaming group (Group 3), VR fishing group (Group 4), and VR watering group (Group 5). Groups 1 and 2 compared the effects of environment types, and Groups 2 to 5 compared the effects of complexity of interaction. Each group experienced a 10-min interactive VR scenario, assessed the intervention effectiveness by subjective scales and physiological measures, and captured the scenario experience through post-intervention interviews.

#### 3.1.3. Materials and Instruments

##### VR Scenarios

Based on Study 1, Study 2 further distinguished the type of environment and the complexity of interaction in the interactive VR scenarios, which may influence the restorative environment, presence, changes in anxiety and depression, and the scenario experience, as shown in Figure 6.

To compare the type of restorative environment, a VR City environment (Group 1) was developed to contrast with the park environment (Group 2) from Study 1 on the basis of Windridge City (https://assetstore.unity.com/packages/3d/environments/roadways/windridge-city-132222, accessed on 17 December 2019) via Unity 5.6.4 (Unity Technologies Inc., San Francisco, CA, USA).

The complexity of interaction was defined as the cognitive and physical effort required by participants to complete VR interactive activities, and its operationalization was defined as the number of interactive tools, limb movement amplitude, range of motion, and degree of whole-body coordination and cooperation. To compare the effects of handle operation in interaction, immersive VR environments could be developed in two models: (1) automatic viewing model (Group 2): the user experienced the scenarios played automatically by the system on a set route, without handles or interactive activities; (2) free-roaming model (Group 3–5): participants could freely move within a certain range and use the handle to perform interactive actions. To further compare the impact of the complexity of interaction, the activities were categorized into easy, medium, and hard levels. In the easy level (Group 3), participants only needed to move by using the main handle in the right hand (for directional control) and holding the secondary handle in the left hand (to assist in completing the activity). At the medium level (Group 5), the participants operated the handle with both hands to perform the watering action whose activation process was relatively smooth. Participants needed to pull the trigger button to pick up the kettle by the secondary handle, moving by the main handle when watering. As the body crouches to lean forward, the water will flow out of the kettle, and the garden area will grow vegetables automatically. In the hard level (Group 4), participants were expected to perform a complex action of fishing whose process was relatively tedious and the range of movement was large. Participants needed to pull the trigger button on the secondary handle to pick up the fishing rod and move through the main handle to cast the fish bladder into the lake. The handle vibrated to indicate that the fish was hooked, then they pulled up the fishing rod, touched the caught fish with the main handle after the fish had fallen onto the lawn, put down the fishing rod to pick up the fish, and put it into the fish basket.

##### Subjective Scales

The subjective scales were still measured using the Restorative Environment Scale (RES), the Presence Questionnaire (PQ), the State Anxiety Scale (SAI), and the Self-rating Depression Scale (SDS) as Study 1.

##### Interview

A structured interview was conducted after the scenario experience. Then the interview recordings would be analyzed based on grounded theory. Grounded theory is proposed by sociologist Barney Glaser and his partner Anselm Strauss in 1965, which is a research method that summarizes the collected data inductively, and is a bottom-up construction of substantive theory by the qualitative research method [88,89]. The techniques used to analyze and classify primary source materials include three levels: open coding, axial coding, and selective coding.

##### Instruments

A set of BIOPAC Systems, Inc MP160 biophysical measurements were used to record EEG and EMG before and during the VR experience. EEG signals were acquired using an EEG100C amplifier, 2 LEAD110 shielded leads, 1 LEAD100 unshielded lead, and 3 disposable patch electrodes; EMG signals were acquired using an EMG100C amplifier, 2 LEAD110S shielded leads, 1 LEAD100A unshielded lead, and 3 disposable patch electrodes.

The VR equipment was the same as in Study 1. The host device was a set of Aliens (CPU: Intel ® Core ™ i7-8700K CPU @ 3.70GHz 3.70GHz; GPU: NVIDIA GeForce GTX 1080 Ti) for optimal definition and smoothness.

#### 3.1.4. Experimental Procedures

The experimental procedures included the following steps:

(1) Pre-test for anxiety and depression: participants took the SAI and SDS surveys and were selected when their scores were both between 40 and 60.

(2) Baseline acquisition of EEG and EMG: after signing the informed consent, physiological parameters were taken for 5 min. First, EMG acquisition: first wipe the skin surface with alcohol and saline, then paste the disposable patch electrodes and connect the electrode wires, VIN+ (input) and VIN- (output) at the flexor muscle of the right forearm, GND at the elbow, away from the VIN+ and VIN-. Secondly, EEG acquisition: first wipe the skin surface with alcohol and saline, then paste the disposable patch electrodes and connect electrode wires, with VIN+ and VIN- sites Fp1 and Fp2 with reference to the international 10–20 standard electrode positions, GND at the mastoid behind the right ear.

(3) VR adaptive training: participants of Groups 3–5 were trained, which matched the section in Study 1.

(4) VR experience: participants were randomly assigned to one of five groups and experienced the VR scenario for about 10 min. EEG and EMG were captured during the VR scenario experience. However, the experiment was terminated immediately if VR motion sickness occurred and could not be relieved.

(5) VR scenario experience interview: after removing the VR equipment and physiological instruments, participants would be asked questions about “How did you feel in the VR scenario, what was it specifically?”.

(6) Post-test: participants completed the RES, PQ, SAI, and SDS.

### 3.2. Results

#### 3.2.1. The Restorative Environment

SPSS 25 was used for statistical analysis, the restorative environment scores conformed to a normal distribution by the K-S test. The descriptive statistics of the total score and their dimensional scores are shown in Figure 7. The one-way ANOVA showed that there were no group differences among the five groups, *F(4,164) = 0.348, p = 0.845*.

#### 3.2.2. Presence

The results of the descriptive statistics of the total score and its dimensional scores are shown in Figure 8. A one-way ANOVA revealed that there was a significant difference in presence among the five groups, *F(4,174) = 4.093, p = 0.003*. A further LSD post hoc test revealed that there was a significant difference between Group 1 and Group 3, *t(1) = 3.513, p = 0.001*; the difference between Group 2 and Group 3 was significant, *t(1) = 2.946, p = 0.004*; the difference between Group 3 and Group 4 was significant, *t(1) = −2.767, p = 0.006*; the difference between Group 3 and Group 5 was significant, *t(1) = −3.327, p = 0.001*.

#### 3.2.3. EEG and EMG

Pre-processing of EEG data by Acqknowledge 4.2.1: (1) high pass low pass filtering: select digital filters/IIR/band pass low+high option under transform menu, then enter 1 and 40 Hz in low freq and high freq respectively; (2) comb filtering: select digital filters/com band stop option under transform menu, then enter 50 Hz in base frequency; (3) artificial artifacts removal: click on the horizontal coordinates of the graph and set the timescale to 2 s, and remove cliff-changing curves to prevent distractions caused by eye movements, large head, and body movements and sweating, etc.; (4) Matlab artifacts removal: firstly, detect eye movement artifacts by sliding window function peak-to-peak threshold method, move forward in a time window of 1000 ms in each trial, then it was marked as eye movement artifacts and excluded if the amplitude change exceeds 150 μV; secondly, exclude drift and other artifacts by a circular algorithm if the absolute value of the data exceeds 100 μV. Then, the relative power spectral densities of α and β waves were calculated via Matlab R2020a.

Pre-processing of EMG data by Acqknowledge 4.2.1: (1) high pass low pass filtering: select the digital filters/IIR/band pass low+high option under the transform menu, then enter 1 and 500 Hz in low freq and high freq respectively; (2) comb filtering: select the digital filters/com band stop option under the transform menu, and then input 50 Hz in base frequency; (3) manual artifact removal: click the horizontal coordinate of the graph and set the timescale to 120 s, and remove the curve with amplitude over 6 mV; (4) Wavelet denoising by Matlab Wavelet Toolbox: choose sym8 wavelet base with better noise elimination effect and shorter operation time and carry out 5-layer wavelet packet decomposition, choose fixed form threshold method for wavelet reconstruction. Then, the MF of EMG was calculated via Matlab R2020a.

The EEG and EMG data were analyzed by SPSS 25 after the K–S test conforming to a normal distribution. The descriptive results of changes in the relative power spectral density ratios of α and β waves ($V_{\alpha /\beta }=\alpha /\beta_{pre-test}-\alpha /\beta_{in-test},V_{\alpha }=\alpha_{pre-test}-\alpha_{in-test},V_{\beta }=\beta_{pre-test}-\beta_{in-test}$) and the changes in median frequency of EMG ($V_{MF},V_{MF}=MF_{pre-test}-MF_{in-test}$) are shown in Figure 9 and Figure 10.

$V_{\alpha /\beta }$ met the homogeneity of variances test and had a significant difference among groups from the one-way ANOVA, *F(4,138) = 3.337, p = 0.012*. Further LSD post hoc tests revealed that there was a significant difference between Group 2 and Group 5, *t(1) = 3.446, p = 0.001*; and a significant difference between Group 4 and Group 5, *t(1) = 2.530, p = 0.013*. In addition, $V_{MF}$ met the homogeneity of variances test but there was no difference among groups, *F(4,161) = 0.140, p = 0.967*.

Descriptive statistics for pre-test ($SAI_{0},SDS_{0}$), post-test ($SAI_{1},SDS_{1}$), and changes of state anxiety and depression ($V_{SAI}=SAI_{0}-SAI_{1},V_{SDS}=SDS_{0}-SDS_{1}$ ) are shown in Figure 11.

A one-way ANOVA showed that there was no significant difference of the $SAI_{0}$ among five groups, *F(4,157) = 1.896, p = 0.114*; there was no significant difference of the $SDS_{0}$ among five groups, *F(4,157) = 0.993, p = 0.413*.

A paired *t*-test showed that there was a difference between pre-test and post-test. For Group1, there was a significant difference between $SAI_{0}$ and $SAI_{1}$, *t(29) = 5.040, p < 0.001*, but no significant difference between $SDS_{0}$ and $SDS_{1}$, *t(29) = 0.349, p = 0.730*. For Group 2, there was no significant difference between $SAI_{0}$ and $SAI_{1}$, *t(32) = 0.521, p = 0.606*, and no significant difference between $SDS_{0}$ and $SDS_{1}$, *t(32) = 0.121, p = 0.905*. For Group 3, there was a significant difference between $SAI_{0}$ and $SAI_{1}$, *t(29) = 2.272, p = 0.031*, but no significant difference between $SDS_{0}$ and $SDS_{1}$, *t(29) = 1.515, p = 0.141*. For Group 4, there was no significant difference between $SAI_{0}$ and $SAI_{1}$, *t(36) = 1.457, p = 0.154*, and no significant difference between $SDS_{0}$ and $SDS_{1}$, *t(36) = 1.800, p = 0.080*. For Group 5, there was a significant difference between $SAI_{0}$ and $SAI_{1}$, *t(31) = 3.777, p = 0.001*, and a significant difference between $SDS_{0}$ and $SDS_{1}$, *t(31) = 2.639, p = 0.013*.

A one-way ANOVA showed that $V_{SAI}$ met the homogeneity of variances test and had a significant difference among groups, *F(4,157) = 2.773, p = 0.029*. A further LSD post hoc test revealed the difference between Group 1 and Group 2 was significant, *t(1) = 2.942, p = 0.004*; the difference between Group 1 and Group 4 was significant, *t(1) = 2.405, p = 0.017*; the difference between Group 2 and Grou p5 was significant, *t(1) = −2.093, p = 0.038*. However, $V_{SDS}$ met the homogeneity of variances test, the scores were not significantly different among the groups, *F(4,157) = 0.746, p = 0.562*.

#### 3.2.4. Correlation Analysis

Pearson correlation analysis of the RES, PQ, $V_{SAI},V_{SDS},V_{\alpha /\beta }$, and $V_{MF}$ is shown in Table 2. The RES and PQ were significantly positively correlated, r = 0.422, *p* < 0.001, the RES and $V_{SAI}$ were significantly negatively correlated, r = −0.240, *p* = 0.003, the RES and $V_{SDS}$ were significantly negatively correlated, r = −0.172, *p* = 0.036. PQ, and $V_{SAI}$ were significantly negatively correlated, r = −0.238, *p* = 0.003, PQ and $V_{SDS}$ were significantly negatively correlated, r = −0.259, *p* = 0.001, $V_{SAI}$ and $V_{SDS}$ were significantly positively correlated, r = 0.185, *p* = 0.020.

#### 3.2.5. Mediation Effects

The relationship between the RES, PQ,$V_{SAI},V_{SDS}$ was further explored through linear regression, using the Bootstrap mediation effect test by the SPSS Process V3.5 which indicates a significant mediation effect if the 95% confidence interval does not contain 0 (Hayes 2017). Using the RES as the independent variable, PQ as the mediating variable, and $V_{SAI}$ and $V_{SDS}$ as the dependent variables, Model 4 in the process was used with a sampling size of 5000, and the results showed a significant total effect of the RES on *$V_{SAI}$, β = −1.904, p = 0.003, 95% Cl (−3.156,−0.651)*, a direct effect that was not significant, *β = −1.355, p = 0.051, 95% Cl (−2.721,0.010)*, and an indirect effect was significant, *β = −0.548, 95% Cl (−1.155,−0.019)*, i.e., the PQ played a mediated role in the effect of the RES on $V_{SAI}$, with a mediation rate of 28.80%. The total effect of the RES on $V_{SDS}$ was significant, *β = −1.076, p = 0.036, 95% Cl (−2.079,−0.073)*, the direct effect was not significant, *β = −0.370, p = 0.500, 95% Cl (−1.451,0.712)*, and the indirect effect was significant, *β = −0.706, 95% Cl (−1.308, −0.188)*, i.e., the PQ played a mediated role in the effect of the RES on $V_{SDS}$, with a 65.63% mediation rate.

#### 3.2.6. Scenario Experience

Nvivo 12 was used to code the VR scenario experience. The open coding included 23 nodes, e.g., “Reality”, 64 cases, e.g., “The roads were realistic when turning”. The axial coding included 6 nodes, like "presence". The selective coding included "good experience" and "bad experience", as shown in Appendix A Table A2. The coded reference points were used to analyze the proportion of experiences. Among the good experiences, comfort and relaxation accounted for the highest percentage, 31.17%, followed by novelty and interesting, 26.62%, and satisfaction, 23.38%; among the bad experiences, motion sickness accounted for the highest percentage, 27.01%, followed by the rough model, 12.64%, followed by less interaction and freedom, 12.07%.

### 3.3. Discussion

#### 3.3.1. The Restorative Environment in the VR Scenarios

The restorative environment results showed that the VR scenarios in the five groups had good restoration and there were no between-group differences. As far as the environment type was concerned, the restoration of the urban environment was higher but not significantly more than the park environment. Specifically, the fascination and compatibility of the urban environment were better, but the being away aspect of the park environment was better. Contrary to expectation, the urban environment was well restorative while a loss of positive experience occurred in the park. The urban environment in this study was an urban street environment, including primarily architectures with the same blue sky, trees (a few scattered on the roads), and viewing road routes as in the park environment. Perhaps the natural elements in the city played a restorative role, but it could not be denied that the urban environment in VR had restorative characters. Importantly, a sense of security was essential for a restorative environment [90,91], which might also be suitable for VR scenarios. No matter what kind of VR environment it is, as long as the danger was kept away and security was satisfied, it could help produce restoration. What influenced the restoration in the park might be the intensity of the environmental elements, such as the bright colors and the high light saturation as evidenced by participants’ interviews. Research has proven that the brightness level is associated with people’s perception of VR natural environment characteristics, for creating a sense of security and reducing anxiety, and medium brightness is more helpful to improve state anxiety than the overly bright [92]. However, the park environment made it easier to leave the urban context to restore physical and mental state, which was in line with Kaplan’s focus on the being away and resting directed attention of the natural environment [38]. Moreover, the environmental elements in the park were favored by participants, as reported in the scenario experience interviews, the house, garden, vegetables, and so on made them comfortable and relaxed, helping them enjoy the virtual environment. These highlighted the importance of environment settings for restoration.

As far as the complexity of interaction, there were some special findings in the free-roaming and watering scenarios. For the free-roaming scenario, the group got the lowest fascination and compatibility but highest extent in the interactive VR scenarios. Although participants were allowed to move freely during the free-roaming scenario, their behavior was slightly restricted. For example, to avoid motion sickness, the speed of movement was set to slow and constant, and to avoid collisions, it was not possible to move to areas with many trees. It seemed that a certain amount of friendly interactivity was sacrificed, the slightly restricted interaction enhanced by perceiving the richness of the natural environment. For the watering scenario, the highest restoration was observed in comparison to the fishing, free-roaming, and automatic viewing scenarios. Specifically, the being away and fascination and compatibility were better in the watering scenario than in the fishing and free-roaming scenarios. These suggested making the complexity of the interaction easy for people to learn and adapt behavioral habits may be an important factor in assessing VR restorative environments. More specifically, appropriate interactive activities could enhance the restoration. From an ecological perception perspective, affordance theory emphasizes the complementarity between the environment and people. The environment provides people with settings that have specific functions, and people perceive environmental information and behave correspondingly [93]. The same was true in interactive VR environment design. The park environment not only presented natural elements, but also provided functions such as exploration, rest, and pleasure, which were detailed in specific contexts, such as watering in the garden or fishing at the lake, and people perceived the restoration of the environment in the interactive behaviors. Moreover, the ability of the VR environment to provide appropriate information and people’s fluent behaviors might be crucial to the restoration.

#### 3.3.2. The Presence in the VR Scenarios

The presence results showed that the VR scenarios in the five groups had good presence and there were between-group differences. In terms of the environment type, the urban environment scored higher on sensory fidelity and immersion than the park environment, which pointed out the importance of the simulation fidelity and quality of the environmental models to the presence. In addition, the sense of connection to living being inspired by the environment influenced the presence in the interviews. Participants wanted to see people or animals on the street, or else wanted to escape the environment during the long-time experience in the urban environment, but few participants reported isolation in the park environment. Sociologist Louis Wirth points out that demographic variables (size, density, and heterogeneity) shape the specific state and way of life in a city [94]. It might suggest that social characteristics are prominent in urban environments and people formed crowd-driven impressions of the city. Researchers find that companionship, rather than being alone, enhances people’s taste for urban environments [95,96], which is linked to the need for security and social needs. The psychological needs stimulated by the VR environment were important for presence. Moreover, different contexts correspond to specific psychological characteristics. Social presence theory emphasizes the sense of being with another in a social context [97]. The biophilia hypothesis points to emotional connection and positive response to the nature context [98,99]. These were reminders of the importance of specific cognitive experiences and emotional needs for the VR scenarios to enhance presence.

In terms of the complexity of interactive activities, the free-roaming activity scored significantly lower than the other groups, suggesting that the presence depends heavily on the matching of environment and interaction. Firstly, the interactive activity that did not match the behavior negatively affects the experience. As seen in the results, the automatic viewing group had a higher presence (especially the adaptation/immersion and interface quality) than the free-roaming group, despite the fact that they both experienced the same environmental elements. The handle interaction in the free-roaming group had a negative effect on the presence, as it moved at an inappropriate speed, creating limitations on the participant’s ability to explore the environment. Secondly, the presence could be more wonderful in the interactive scenarios than those without interactive activities when the interactions were not designed too complex to cause cognitive load. It was proven by the watering activity that got the highest presence of the interactive VR scenarios. On the one hand, compared to automatic viewing, the watering scenario got higher involvement and adaptation/immersion. Well-designed interactions enhanced the presence in the same environmental conditions. On the other hand, the involvement in the fishing scenario was lower than in the watering scenario, suggesting that complex interactive activities could reduce the positive experience and even make it difficult and fatiguing for participants, as described in the interviews. Interaction, as one of the unique advantages of virtual reality, plays an important role in enhancing the experience of presence [100]. As for usability goals in interaction design, interaction needs to ensure the effectiveness, efficiency, safety, utility, ease of learning, and remembering [101]. The interactive usability in VR scenarios could better convey environmental information from the spatial dimension and enhance the connection between the environment and people, but when and where to engage in what kind of interaction in VR intervention contexts, and the physical and mental changes caused by interaction still need to be studied in depth.

#### 3.3.3. EEG and EMG for VR Psychological Intervention

The EEG results showed that the level of brain fatigue had been improved for the positive values of $V_{\alpha /\beta }$ in all groups. From the environment, the park environment was more brain-friendly than the urban environment for the higher $V_{\alpha /\beta }$. Specifically, participants had higher α power but lower β power in the urban environment, making it difficult to promote a focused but relaxed state. From the complexity of interaction, though the lowest level of brain fatigue release was found in the $V_{\alpha /\beta }$ of watering scenario, further analysis revealed that the watering scenario activated both high α and β power, implying a very pleasant and efficient state. This evidence profoundly exposed the intervention effect of appropriate VR interactions on physical health in two ways. Firstly, the $V_{\alpha /\beta }$ in the watering activity group was significantly lower than that in the park environment group, and the park environment group got higher β power than the free-roaming, fishing, and watering groups. The level of brain activation in the automatic viewing scenario was higher than in the interactive scenarios, which was not as expected, originally the β power would increase as the interaction becomes more complex. A possible reason for this was that attention enhanced people’s perception of the environment, but also created a sense of tension. In addition, interactive activities might be beneficial in developing a relaxing VR environment and alleviating excessive attention and tension for people with anxiety and depression. Secondly, though the $V_{\alpha /\beta }$ of the watering scenario was significantly lower than that of the fishing scenario, the fishing scenario got higher α and β power, indicating that, although the interactive activities promoted relaxation and brain fatigue, the complex interactive activities had a negative potential impact on intervention.

The EMG results showed that the median frequency increased in all groups during the experiment. There were no differences among groups and no significant muscle fatigue was imposed, indicating that the scenarios were designed to be friendly and free of cognitive load.

The EEG and EMG results were consistent with the restorative environment and presence in the VR scenarios, implying validation both physiologically and psychologically. EEG and EMG provide an opportunity to track the attention and cognitive load, helping measure the changes of anxiety and depression during the tasks, which is also confirmed by relevant studies [102,103,104]. EEG and EMG might quantify the experience and mechanism of VR interactive scenarios, and future research could further explore their role in providing a theoretical basis for VR interventions in anxiety and depression.

#### 3.3.4. Changes in Anxiety and Depression and Effect Factors

The results showed that all groups experienced a reduction in anxiety and depression, proving the possibility of VR intervention. For anxiety, there were significant intervention effects on state anxiety in the urban environment group, free-roaming group, and watering group, suggesting that the urban scenario and interactive scenarios helped to alleviate state anxiety. This supported the importance of environments and interaction in improving anxiety and depression. On the environment type, the urban environment had significantly better intervention effects than the park environment, which was inconsistent with existing research findings. Natural environments can stimulate more positive physical and psychological responses that are more conducive to alleviating negative emotions, eliminating fatigue and enhancing health [52,105,106]. While urban green spaces can stimulate the same restorative potential and even rehabilitative effects as nature [107,108]. Perhaps the natural elements as well as the high scores of the restorative environment and the presence of the city played an intervention role. More importantly, it highlighted the possibility of setting up diverse and restorative VR environments, which could be natural, familiar, or even imaginative to the spiritual world. On the interaction, a significant reduction in state anxiety was discovered in the free-roaming scenario and watering scenario, proving that simple and moderately difficult interactive activities helped improve state anxiety. The importance of appropriate interaction for enhancing restoration was also proved by the watering group, which was significantly more effective than the park environment group in intervening with state anxiety. Moreover, while interactive activities enriched intervention, complex interactions could have negative effects, which was supported by the result that the urban scenario was significantly more effective than the fishing scenario. Optimal interaction stimulated the best experience.

For depression, only the watering scenario showed a significant intervention effect on it, which differed from the results of Study 1 in which only depression was significantly reduced in participants with mild to moderate anxiety and depression after experiencing four interventions. This might be due to the fact that only scenarios with appropriate interaction could quickly reduce depression. As stated in self-perception theory, attitudes towards the self are inferred through the individual’s behavior [109]. The negative emotions and thoughts about the self might be improved when VR provides suitable scenarios and the individual’s behavior was positively responded to by the environment. Whereas frustrating behavior might lead to doubt and apprehension about the self, playing a negative role in improving anxiety and depression.

Factors affecting the intervention effects of anxiety and depression were further verified in the correlation and mediation effect analysis, where the restorative environment was significantly positively correlated with post-test anxiety and depression, respectively, and presence was significantly positively correlated with post-test anxiety and depression. Moreover, presence mediated the effect of the restorative environment on anxiety and depression, suggesting that presence played an important role in influencing the intervention, and participants’ psychological experiences directly influence improving physical and mental states by VR. This supported the finding that the level of presence influenced emotional states in VR [69,110]. Being in a restorative environment and engaging in physical activities contribute to the long-term relief of negative emotions and a sense of well-being [111]. However, there might have been other factors influencing the effect of the intervention that warranted further research. For example, researchers have pointed out that the consistency between environmental preference and environmental type affects the restoration effect. The physical and mental restoration effect would be better if the individual environmental preference is consistent with the environment experienced. For instance, people who prefer nature have a high consistency effect when experiencing the natural environment, which is easy to stimulate the recovery potential and positive emotions [107,108,112]. In general, VR psychological intervention scenarios should be designed based on the psychological need, and cognitive and behavioral characteristics of the intervened person, creating restorative environments and appropriate interaction to stimulate a sense of presence in order to reduce negative emotions and thinking.

#### 3.3.5. VR Scenario Experience and Optimization

The qualitative analysis revealed the scenario experience during the intervention. In terms of good experience, the VR scenarios were applauded for the relaxing, safe, and natural interactions, leading to a sense of presence (realism and absence of time), enjoyment (comfort and relaxed, novelty and interesting, satisfaction), and even sublimation (attachment and contemplation). The good experience promotes participants to immerse themselves in VR, feel the beauty and fun of the virtual environments, accept the challenge and obtain the power of control which improve anxiety and depression. However, the bad experience illuminated where the VR scenarios could be improved in terms of setting, interaction, and equipment. In terms of setting, it was important to (1) design the scenario size in relation to the length of the experience and speed of movement, (2) provide a rich variety of scenarios elements that vary across paths, (3) design the lighting and color of the scenarios in relation to perceptual properties and attentional preferences, and (4) focus on the quality of the model, the height of the location, and add rigid bodies and colliders in order to detect collisions. In terms of the interaction, it was important to (1) show positioning and purpose by clear signage and clear interaction tasks and (2) provide interesting and appropriate interactions with multisensory channels. In terms of equipment, issues such as disfluency, low resolution, and motion sickness stood out. For the disfluency and low-resolution issues, on one hand, preliminary tests should be carried out when designing VR scenarios to avoid scene overload and technical limitations. On the other hand, it is best to use high-performance hosting equipment and displays. For motion sickness issues, motion sickness seems prevalent in VR and is mainly manifested as a certain sense of dizziness, headache, nausea, and fatigue [113,114], which is related to the inconsistency between the visual state and the movement state. The reasons for this were explained as the brain perceiving virtual movement but no movement in the real body, and on the other hand, the delay in the image and the excessive speed of movement in interactive VR scenarios could also cause undesirable feelings. Studies have shown that motion sickness affects people’s emotional, cognitive, and behavioral performance in VR [115,116]. However, there were significant individual differences in motion sickness and most participants reacted only slightly and were able to adapt quickly. Researchers could avoid the negative effects of motion sickness by designing scenarios that match the realistic movement and enhance a good experience.

In general, the experience in various VR scenarios was diverse, and it is not brought by a single factor. It was particularly important that scenario setting, interaction, and device were fully integrated, forming a coherent whole that provided a good experience to help relieve anxiety and depression.

#### 3.3.6. Limitations

There were some shortcomings in this study, such as the sample size being small, and only the effects of restorative and present VR on anxiety and depression mood were explored; the intervention on anxiety and depression disorder need to be further studied. Moreover, the problems with the VR scenarios in setting, interaction, and equipment from Appendix A need to be improved. In addition, the EEG and EMG were partially disturbed during the VR interaction activities, for example, the head-mounted displays and interaction activities made individual participants sweat easily, with some accuracy loss.

## 4. Conclusions

These studies constructed a VR urban environment and a VR park environment which contained five interactive scenarios such as free-roaming activity in the park and flying a kite in the lawn area, watering vegetables in the horticultural planting area, fishing in the water area, and feeding birds in the forest area. These were developed and repeatedly modified based on landscape art design, interaction design, motion capture, and VR testing, corresponding in turn to interaction with or without handles, small upper body movements, large body forward leaning movements, large body backward leaning movements, and small body crouching movements might reveal the complexity of VR interaction activities, which play a central role in influencing participants’ VR experiences and interventions. Furthermore, subject scales, EEG and EMG, and scenario experience were used to explore the intervention effects of VR scenarios for people with mild to moderate anxiety and depression. The main findings were as follows:

(1) Restorative and present VR scenarios were effective in alleviating state anxiety and depression.

(2) The restorative environment and presence were significantly and positively related to the reduction of anxiety and depression respectively; presence mediated the restorative environment on the recovery from anxiety and depression.

(3) Environmental settings (types and elements, etc.) and the complexity of interaction, human factors (needs, attention preferences, behavioral characteristics), and maturity of VR devices and technology were key factors influencing the scenario experience and improving anxiety and depression. When the VR environment and interaction were well-designed, it helped to intervene with anxiety and depression, otherwise, an inappropriate design could trigger frustration. Furthermore, the core mechanisms of the VR intervention of anxiety and depression call for more research.

VR technology has natural advantages in enhancing mental health, with good ecological validity in scenario presentation, data collection, assessment and intervention of physical and mental states, etc. However, the rich restorative environments and natural interactions need to be further explored, especially not only to create natural and urban environments but also to provide imaginative scenarios which people really need. Moreover, physical and mental states in interactive VR scenarios would be assessed, building multimodal indicators and rich comprehensive models of humans, machines, and environments [117] to reconceptualize and reanalyze humans. Future research could consider designing interesting and appropriate VR interactive scenarios in an ergonomic way, tracking physical and mental mechanisms in the virtual and real world, and creating quality human–computer interaction experiences to enhance the public mental health and well-being.

## Figures and Tables

**Figure 1 ijerph-19-07878-f001:**
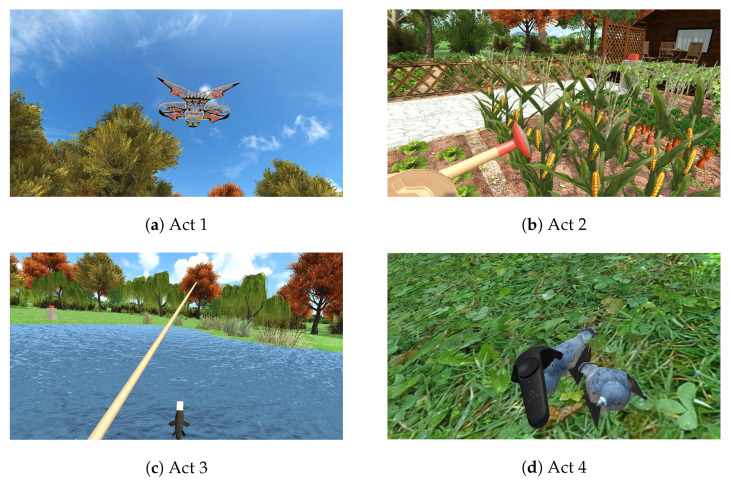
Interactive VR Scenarios in Study 1, (**a**) flying a kite in the lawn area, (**b**) watering vegetables in the horticultural planting area, (**c**) fishing in the water area, and (**d**) feeding birds in the forest area.

**Figure 2 ijerph-19-07878-f002:**
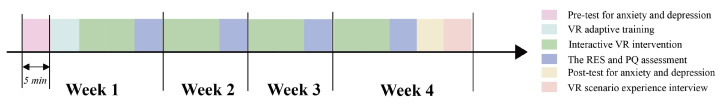
The experimental procedures in Study 1.

**Figure 3 ijerph-19-07878-f003:**
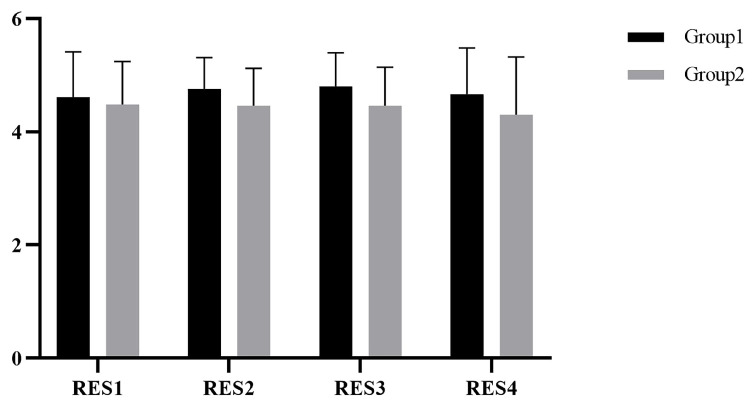
The scores of the restorative environment in Study 1.

**Figure 4 ijerph-19-07878-f004:**
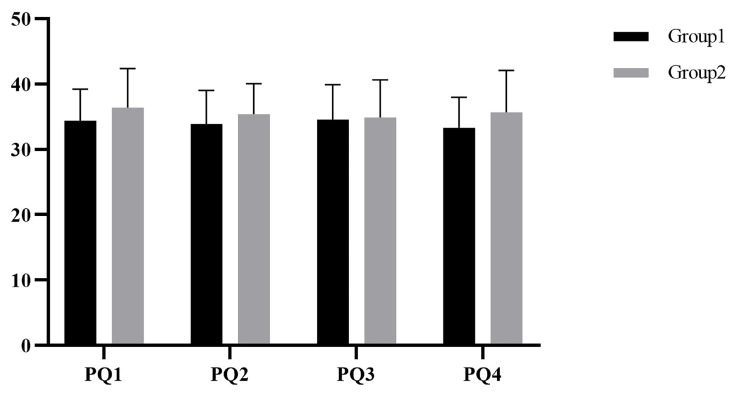
The scores of the presence in Study 1.

**Figure 5 ijerph-19-07878-f005:**
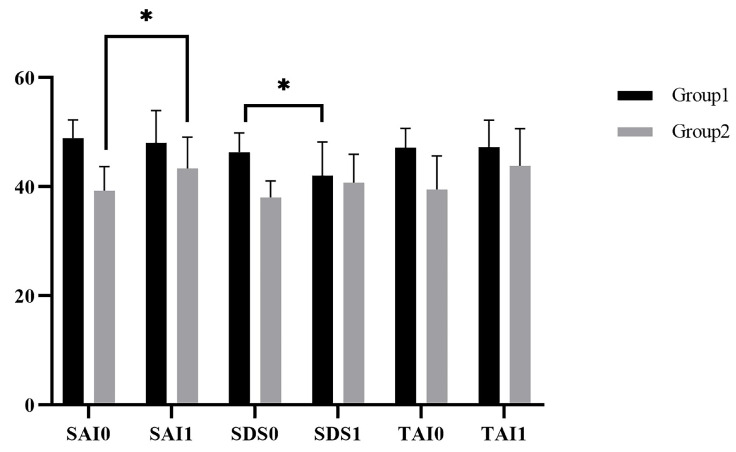
The scores of the anxiety and depression in Study 1. * *p* < 0.05.

**Figure 6 ijerph-19-07878-f006:**
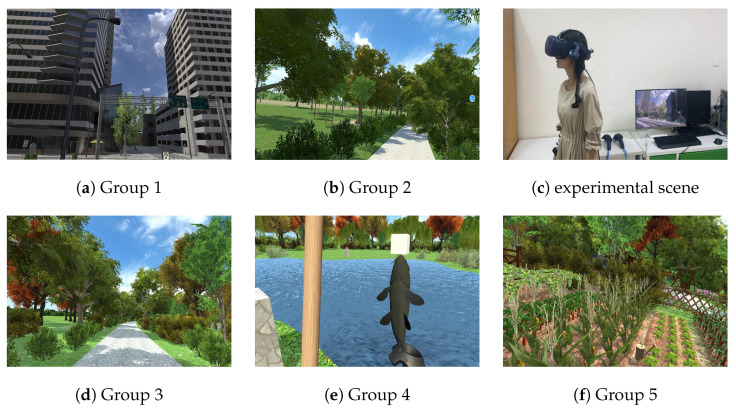
VR scenarios in Study 2, (**a**) automatic viewing the urban environment, (**b**) automatic viewing the park environment, (**c**) experimental scene, (**d**) free-roaming in the park environment, (**e**) fishing in the park environment, and (**f**) watering in the park environment.

**Figure 7 ijerph-19-07878-f007:**
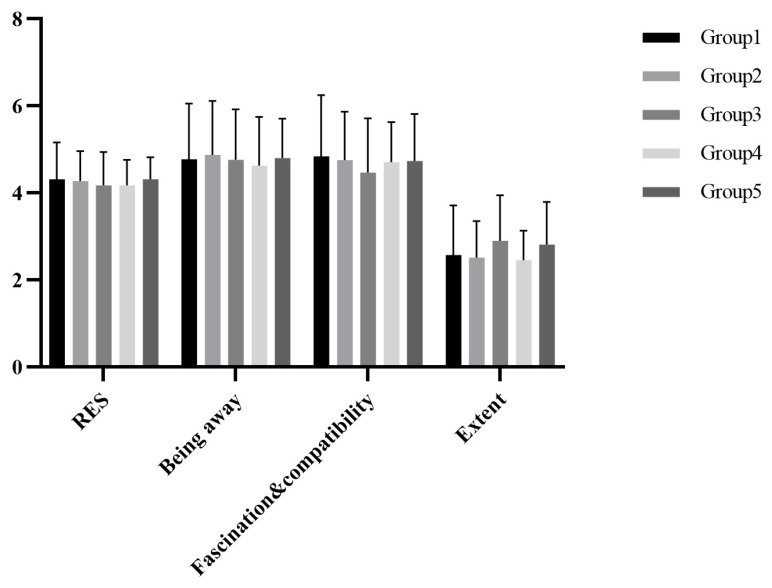
The scores of restorative environment in Study 2.

**Figure 8 ijerph-19-07878-f008:**
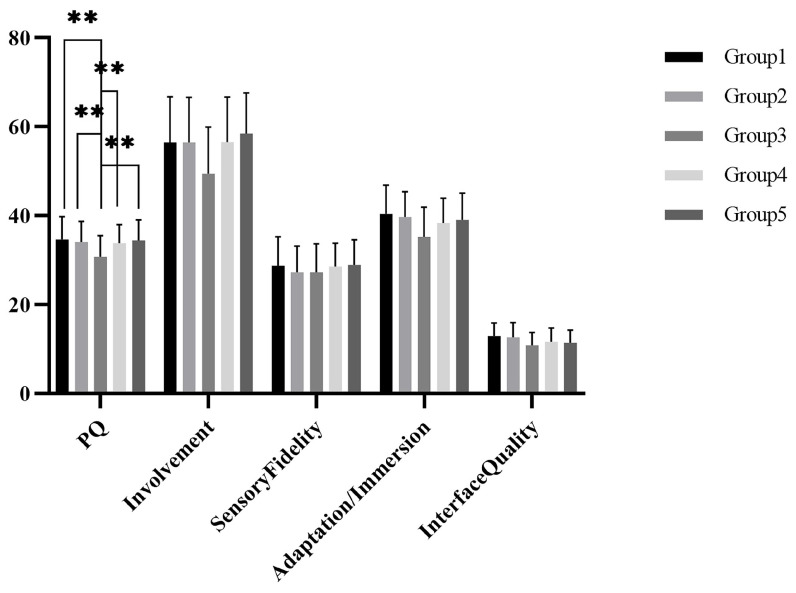
Thescores of presence in Study 2. ** *p* < 0.01.

**Figure 9 ijerph-19-07878-f009:**
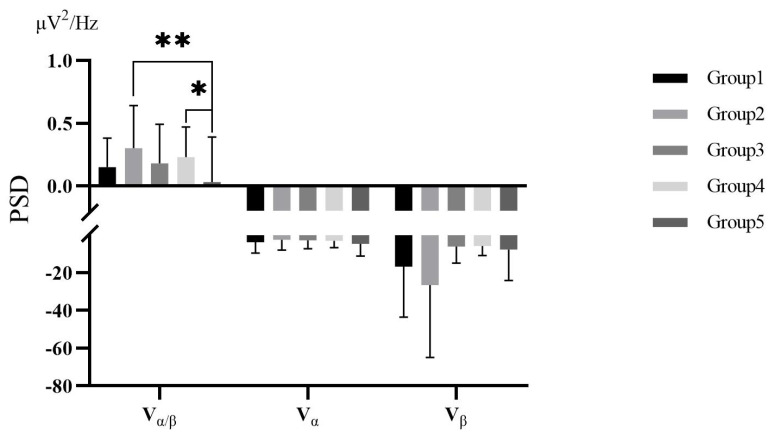
The EEG in Study 2. * *p* < 0.05, ** *p* < 0.01.

**Figure 10 ijerph-19-07878-f010:**
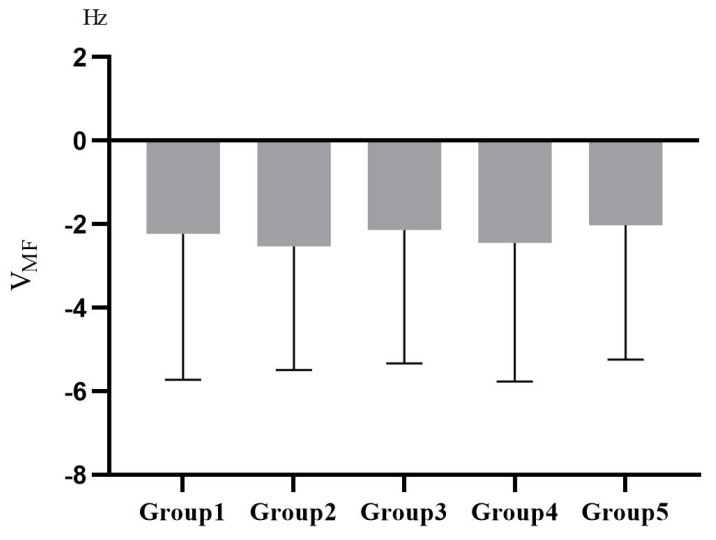
The EMG in Study 2.

**Figure 11 ijerph-19-07878-f011:**
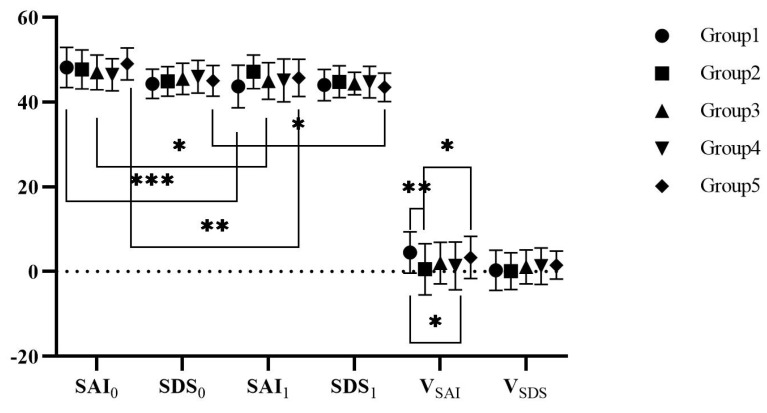
The scores of anxiety and depression in Study 2. * *p* < 0.05, ** *p* < 0.01, *** *p* < 0.001.

**Table 1 ijerph-19-07878-t001:** Basic participant information in Study 2.

	Total		Male		Female	
	Number	Age	Number	Age	Number	Age
Group 1	39	20.23 ± 2.29	15	21.13 ± 2.53	24	19.67 ± 1.99
Group 2	35	20.34 ± 2.98	7	21.14 ± 4.41	28	20.14 ± 2.58
Group 3	36	19.50 ± 1.99	15	19.27 ± 2.01	21	19.67 ± 2.01
Group 4	38	20.74 ± 2.52	20	20.95 ± 2.68	18	20.50 ± 2.38
Group 5	38	20.61 ± 2.93	19	20.79 ± 3.43	19	20.42 ± 2.43
Total	186	20.29 ± 2.58	76	20.63 ± 2.94	110	20.05 ± 2.28

**Table 2 ijerph-19-07878-t002:** The Pearson correlation analysis of variables in Study 2.

	RES	PQ	$V_{\mathrm{SAI}}$	$V_{\mathrm{SDS}}$	$V_{\alpha /\beta }$
PQ	0.422 **				
$V_{SAI}$	−0.240 **	−0.238 **			
$V_{SDS}$	−0.172 *	−0.259 **	0.185 *		
$V_{\alpha /\beta }$	0.011	0.046	−0.104	0.015	
$V_{MF}$	0.103	0.143	−0.162	−0.144	0.100

* *p* < 0.05, ** *p* < 0.01.

## Data Availability

The raw data for this article were provided by the authors.

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
