# Peer review of "Effects of Restorative Environment and Presence on Anxiety and Depression Based on Interactive Virtual Reality Scenarios†"

_ijerph, 2022, doi:10.3390/ijerph19137878_

Round 1

Reviewer 1 Report

This paper focuses on the effect of restorative environment and presence on anxiety and depression based on interactive virtual reality scenarios, and its work is very valuable. However, I have some comments and suggestions which hopefully would help perfecting the manuscript. My comments and suggestions are as follows.

1. Line 152 mentions that a 2 (anxiety and depression state) * 4 (interactive VR scenarios) experimental design was used, but there is no treatment without VR scenes as a control group, so how do the results prove that VR scenes are beneficial for reducing anxiety and depression?

2. Line 157 mentions that a randomized approach was taken to balance the order effects. So what is the order of experiments in this paper? What are the effects of the different sequences?

3. Line 315 mentions that groups 1 and 2 compared the effects of environment types, and groups 2 to 4 compared the effects of complexity of interaction. So what is the purpose of the group 5?

4. Is it reasonable to use five different scenarios of free-roaming, flying a kite, watering vegetables, fishing, and feeding birds to distinguish the complexity of interaction? Is it possible to control extraneous variables with different difficulty settings in the same scene?

5. Line 728 mentions that environmental settings (types and elements etc.) and the complexity of interaction, human factors (needs, attention preferences, behavioral characteristics), and maturity of VR devices and technology were key factors influencing the scenario experience and improving anxiety and depression. But, for example, is the realism and the immersion of the scene also key factors?

6. Some of the latest research in VR should be summarized and followed up, such as DOI: 10.1080/17538947.2017.1419452;  https://doi.org/10.1111/tgis.12914

Reviewer 2 Report

In the article by  Wang et al,  the authors have studied Effects of
Restorative Environment and Presence on Anxiety and Depression. For
this, they have used Interactive VR scenarios. This includes various
scenarios such as automatic viewing of urban and park environments as
well as free roaming, watering and some other activities in the
virtual environment.
This study is important as it will provide insight into importance of
using virtual reality scenarios to alleviate anxiety and depression to
a certain level.
Manuscript has been written nicely, however I found some parts in the
methodology and discussion part were not clearly explained and were
difficult to understand. I have following comments/suggestions which
will strengthen the manuscript:

1) Line 144-146: “the other (Group 2) based on whether the scores on the
State Anxiety, Trait Anxiety and Depression scales were all greater
than or equal to 40 and less than or equal to 60”. Please explain the
level of anxiety and depression in Group 2. How is it different in
terms of anxiety and depression level from Group1?

2) Please explain why relative power spectral density ratios of α and β
waves was measured in this study. Relevance of the measure to anxiety
and depression should be mentioned in the discussion. Why other
frequency bands were not looked into and compared. Please cite any
study if present in the literature which states that relative power
spectral density ratios of α and β waves  reflect levels of brain
fatigue.

3) It’s not very clear as to which specific electrodes were chosen to
collect EEG activity and on which electrodes changes in α and β waves
was looked into.

4) Line 386-387: “At the same time, EEG and EMG were captured” . At what
time points in the experimental design was EEG data collected. It is
not very clear from the study.  What does authors mean with “pre” and
“post” in  “Vα/β = α/βpre−test − α/βpost−test”. What is the time point
in the experimental design which reflects “pre” and “post”? Please
mention in the methodology

5) Line 618-619: “and no significant muscle fatigue was imposed”. Why do
author wants to check for muscle fatigue in these experiments? Also,
What is the time point in the experimental design which reflects “pre”
and “post”? Please mention in the methodology. Please cite reference
if present in the literature which suggests that the median frequency of
EMG could be a measure for muscle fatigue.

6) Please label y-axis for Vα , Vβ  and also for V_MF

Reviewer 3 Report

The study comes across as innovative and very interesting. The methods are very well implemented and the rigor with which it is conducted places it in a high percentile among current studies. It would be appropriate to indicate before the final discussion a short paragraph where the limitations are stated, such as the small sample.
In addition, it should be specified whether this method results from the study are valid in all types of anxiety/depression, taking into account the fact that depressive comorbidity with psychotic symptoms (although without an overt diagnosis of psychosis) is not so rare. If this has not been established it will have to be added to the limitations.
Overall an excellent study.

Author Response

Thank you so much for the kind encouragement! We have developed the manuscript in the conclusion part.

Round 2

Reviewer 2 Report

Authors have substantially modified the manuscript based on the suggestions. I have few suggestions and if possible authors could include them to improve the manuscript

Comment 1. Regarding point 3 and the response from the authors (pasted below), authors should try to include one or two sentences to show significance of using Fp1 and Fp2 electrodes (for example why not Fz).  

Point 3: It’s not very clear as to which specific electrodes were chosen to collect EEG activity and on which electrodes changes in α and β waves were looked into.

Response 3: Thank you for your comments! This is the EEG data of the forehead recorded by BIOPAC Systems, Inc MP160 biophysical measurements with the positive and negative electrode sites Fp1 and Fp2 with reference to the international 10-20 standard electrode positions. The power spectral densities of α and β  in the forehead and their variations were studied. We modify the methodological section of the study.

Comment 2. Since the EEG was also recorded during the in-test condition, it will be contaminated by movement artifacts. Authors should include this as part of their study limitations.